# Oppositely Charged Pickering Emulsion Co-Stabilized by Chitin Nanoparticles and Fucoidan: Influence of Environmental Stresses on Stability and Antioxidant Activity

**DOI:** 10.3390/foods11131835

**Published:** 2022-06-22

**Authors:** Miao Hu, Xiaoqian Du, Guannan Liu, Yuyang Huang, Zhao Liu, Shukun Sun, Yang Li

**Affiliations:** 1College of Food Science, Northeast Agricultural University, Harbin 150006, China; humiao7890@163.com (M.H.); d1197942226@163.com (X.D.); lgn19935429117@163.com (G.L.); hm2239173531@163.com (Z.L.); 2College of Food Engineering, Harbin University of Commerce, Harbin 150028, China; huangyuyang1979@hotmail.com; 3National Research Center of Soybean Engineering and Technology, Harbin 150023, China; 4Heilongjiang Green Food Science Research Institute, Harbin 150023, China

**Keywords:** Pickering emulsion, chitin nanoparticles, fucoidan, stability, environment stress

## Abstract

Single emulsifiers exhibit varying degrees of restriction in stabilizing emulsions. Oppositely charged chitin nanoparticles and fucoidan complex particles were used as emulsifiers to stabilize a o/w Pickering emulsion and explore its stability and antioxidant activity under different environmental stresses. The results showed that the emulsion with the smallest mean particle size (1.02 μm) and strongest zeta potential (−29.3 mV) was formed at pH 7. Moreover, at this pH, it presented the highest physical stability and antioxidant activity and the lowest emulsion creaming index. The investigation of the effect of temperature on the stability and antioxidant activity of the emulsion revealed that, after freezing/thawing at −20 °C, the emulsion was unstable, the particle size increased, and the stability and antioxidant activity were low. In contrast, the emulsions treated at 25, 37, and 60 °C displayed no significant differences and exhibited high stabilities and antioxidant activities. Additionally, increasing the salt ion concentration further decreased the emulsion stability and antioxidant activity. Particularly, the emulsion with a salt concentration of 500 mM displayed the lowest stability, and stratification occurred after 30 d of storage. The Pickering emulsion remained stable under different environmental stresses expect for at a temperature of −20 °C and 500 mM salt ion concentration.

## 1. Introduction

An emulsion system is typically considered thermodynamically unstable owing to the high surface energy between the two immiscible phases. Emulsifiers play a vital role in designing delivery systems, and numerous emulsifiers are often used to prepare and stabilize delivery systems, such as surface-active proteins, polysaccharides, surfactants, and phospholipids [1,2]. Over a century ago, Ramsden [3] and Pickering [4] showed that similar to surfactants and other amphiphilic molecules, particles can also be used to stabilize emulsions. Such emulsions are known as Pickering emulsions. Such solid-particle-stabilized emulsions are garnering increased attention, owing to their reduced toxicity, lower cost, and simple recovery properties compared to those of conventional surfactants [5,6,7].

Chitin is treated by electrospinning/acidolysis/ultrasound to form rod-shaped chitin nanocrystals (ChNs), with lengths of <1 μm. After treatment, the amorphous area of chitin is destroyed, the crystallinity of the nanocrystals is significantly improved, and the solubility of ChNs is improved. Because of these properties, ChNs serve as effective food-grade Pickering emulsion stabilizers. Tzoumaki, Moschakis, Kiosseoglou, and Biliaderis (2011) [8] proved that ChNs were considerably effective in stabilizing oil-in-water (o/w) emulsions against coalescence over one month. Cho and Oh (2020) [9] discovered that ChNs influenced the foaming ability and foam stability. Jiménez-Saelices, Trongsatitkul, Lourdin, and Capron (2020) [10] reported that the chitin Pickering emulsion could be incorporated into edible hydrophilic films to stabilize the oil.

Fucoidan (Fuc) is an anionic polysaccharide mainly composed of L-fucose and sulfate groups and is found in brown algae and sea cucumbers. This complex polysaccharide exhibits several biological activities and has been used to fabricate delivery systems for bioactive substances [11]. For example, the addition of algal fucoidan prior to homogenization was reported to improve the stability of o/w emulsions containing bovine serum albumin [12]. The ability of F biopolymer nanoparticles to encapsulate and deliver hydrophilic antibiotics has also been reported [13,14].

Each stabilizer has its own applications and limitations. Although surfactants, such as solid particles [15] and biopolymers [16], can stabilize emulsions, they tend to desorb from the droplet interface and cause droplet instabilities, especially under extreme environmental stresses during food manufacture and storage, such as pH and ionic strength changes, heating, and freezing/thawing. Several studies of the use of oppositely charged species to stabilize emulsions have been reported. Meanwhile, related research has been successfully fabricated the electrostatic complexation relevant to food [17,18,19,20]. Notably, ChNs are positively charged, while F is negatively charged and exhibits various biological activities; however, the use of ChNs and fucoidan to prepare emulsion delivery systems has not been widely investigated.

To overcome the limitations of conventional emulsions, we previously fabricated a Pickering emulsion from positively charged ChNs and negatively charged fucoidan [21]. We verified that the as-synthesized Pickering emulsion comprising oppositely charged particles possessed excellent physical and storage stabilities. In the present study, we proceeded to investigate the influence of environmental stresses on the stability of this Pickering emulsion and predicted that the Pickering emulsion would exhibit significant resistance under different environmental stresses, such as pH, heat, and the presence of salts.

## 2. Materials and Methods

Chitin was obtained from Shanghai Yuanye Bio-technology Co., Ltd. (Shanghai, China), while Fuc was purchased from Saien Biological Technology Co., Ltd. (Chengdu, Sichuan, China). Sunflower seed oil was obtained from Beijing Jinshicang Grain and Oil Trading Co., Ltd. (Beijing, China). Nile red, calcofluor white stain dyes, 2,2-diphenyl-1-picrylhydrazyl (DPPH), and 2,2′-azino-bis(3-ethylbenzothiazoline-6-sulfonic acid) (ABTS) were obtained from Sigma Aldrich (St. Louis, MO, USA). All other chemicals were of analytical grade or higher (Beijing Chemical Reagent Co., Beijing, China).

### 2.1. Preparation of the Pickering Emulsion

ChNs were prepared according to our previously reported method Liu et al. (2021) [21]. The ChN–Fuc complex was prepared by mixing 1% (*w*/*v*) ChNs and 1% (*w*/*v*) Fuc at pH 2 and then freeze-drying for further use. The ChN–Fuc complex (10 mg/mL) was dispersed in 0.01 M sodium phosphate buffer solution (PBS, pH 7.0, 25 °C) as the aqueous phase, to which 5% (*v*/*v*) sunflower seed oil (oil phase) was subsequently added as oil phase. The coarse emulsion was prepared by homogenizing the oil–aqueous phase mixture at 10,000 rpm for 2 min (T25, German IKA Group, Staufen im Breisgau, Germany). The coarse emulsion was then further homogenized at 100 MPa for one cycle (FPG12805, Standard Fluid Power Ltd., Warwickshire, UK) to form the o/w Pickering emulsion.

### 2.2. Evaluation of the Pickering Emulsion Stability under Different Environmental Conditions

To evaluate the effect of the pH on the Pickering emulsion stability, a series of emulsions was prepared, and the pH was adjusted to different values (2, 4, 7, and 9) using 1 M HCl or NaOH.

To evaluate the effect of the temperature on the Pickering emulsion stability, the Pickering emulsion was placed in a water bath or refrigerator at different temperatures (−20, 25, 37, and 60 °C).

The influence of salt on the Pickering emulsion stability was examined by adding different amounts of NaCl (0, 50, 100, and 500 mM) to the Pickering emulsion after preparation.

### 2.3. Particle Size and ζ-Potential

The particle size and ζ-potential of the Pickering emulsion were determined using dynamic light scattering (Zetasizer Nano Zs, Malvern Instruments Ltd., Malvern, Worcestershire, UK). The sample was diluted to clear and transparent using 0.01 M PBS (pH 7.0), and the refractive index was set to 1.46 for the particle and 1.33 for the dispersant according to the method of Xiong et al.

### 2.4. Confocal Laser Scanning Microscopy (CLSM)

Approximately 0.1% (*w*/*v*) Nile red was dispersed in isopropanol and passed through a 0.45 μm filter, while calcofluor white stain dye was mixed with KOH at a ratio of 1:1 (*v*/*v*). Next, the Pickering emulsion samples were diluted two-fold with 0.01 M PBS (pH 7). The Nile red (40 μL) and calcofluor white stain dyes (40 μL) were then added in the diluted Pickering emulsion (1 mL) and incubated for 30 min at 25 °C. The stained Pickering emulsion (5 μL) was placed on a glass slide and observed under a CLSM (SP8, Leica, Germany) at excitation wavelengths of 488 and 633 nm.

### 2.5. Cryo-Scanning Electron Microscopy (Cryo-SEM)

The microstructure of the Pickering emulsion was observed using the cryo-SEM method of Xiao, Gonzalez, and Huang (2016) [22]. Briefly, the sample was placed on the copper sample holders and soaked in liquid nitrogen for quick freezing. Samples were frozen and sublimated at −95 °C for 15 min and subsequently sputter-coated. The microstructure was then observed under a cryo-scanning electron microscope at −130 °C.

### 2.6. Physical Stability of the Pickering Emulsion

The physical stability of the Pickering emulsion was measured using the centrifugal separation method. First, the fresh Pickering emulsion was placed into a centrifuge tube, and the initial height (Hint) was measured. Next, the sample was centrifuged at 2500× *g* for 5 min at 25 °C and the height of the creaming layer (Hcen) was measured. The emulsion physical stability (EPS) was calculated using the equation
$$\mathrm{EPS}\;\left(\%\right)=\frac{H_{\mathrm{cen}}}{H_{\mathrm{int}}}\times 100\%$$

### 2.7. Creaming Index (CI) of the Pickering Emulsion

The fresh Pickering emulsion was placed in a colorimetric tube and sealed with a glass stopper and incubated at 25 °C for 1, 3, 5, and 7 days. The heights of the initial Pickering emulsion (HT) and creaming layer (HS) were then measured. The CI was calculated using the equation
$$\mathrm{CI}\;\left(\%\right)=\frac{H_{S}}{H_{T}}\times 100\%$$

### 2.8. Chemical Stability of the Pickering Emulsion

The peroxide value (POV) and thiobarbituric acid reactive substance (TBARS) values of the Pickering emulsion were measured as follows [23,24,25]: For the POV, the fresh Pickering emulsion was held at 50 °C for 10 days to allow oxidation. The sunflower seed oil was extracted using an isooctane: 2-propanol (3:1, *v*/*v*) solution, and the mixture was centrifuged at 1000× *g* for 2 min at 25 °C. The organic phase (0.2 mL) was mixed with 2.8 mL methanol and butanol (2:1, *v*/*v*) mixture; subsequently, 15 μL ammonium thiocyanate (3.94 mol/L) and 15 μL Fe^2+^ solution (0.144 mol/L FeSO_4_ and 0.132 mol/L BaCl_2_) were added to the solution. The afforded mixture was incubated for 20 min in the dark and the absorbance was measured at 510 nm. The POV concentration was calculated using the standard curve of cumene peroxide.

To determine the TBARS value, the Pickering emulsion (0.1 mL) was diluted to 1 mL using 0.01 M PBS (pH 7.0) and mixed with thiobarbituric acid (2 mL). The mixture was incubated for 15 min in boiling water and then cooled; subsequently, the sample was centrifuged at 1000× *g* for 15 min at 25 °C. The supernatant was then collected, and the absorbance was measured at 532 nm. The TBARS concentration was calculated using the standard curve of 1,1,3,3-tetraethoxypropane.

### 2.9. Storage Stability of the Pickering Emulsion

The fresh Pickering emulsion was stored at 25 °C for 30 days, and its particle size were measured according to the methods described in Section 2.3 to determine the storage stability.

### 2.10. Antioxidant Activities of the Pickering Emulsion

The Pickering emulsion (200 μL) was mixed with ABTS work solution (800 μL). The mixture was incubated in the dark for 6 min at 25 °C, and the absorbance was measured at 734 nm. The ABTS^+^ radical scavenging activities (%) were calculated using the equation:$${\mathrm{ABTS}}^{+}\;\left(\%\right)=\left(1-\frac{A_{i}}{A_{0}}\right)\times 100\%$$
where A_i_ and A_0_ are the absorbances of the Pickering emulsion and control, respectively.

The Pickering emulsion (1 mL) was mixed with DPPH ethanol solution (0.12 mM) and incubated in the dark for 30 min at 25 °C. The absorbance of the sample was then measured at 517 nm. The DPPH radical scavenging activities (%) were calculated using the equation:$$\mathrm{DPPH}\;\left(\%\right)=\left(1-\frac{A_{s}-A_{c}}{A_{b}}\right)\times 100\%$$
where A_s_, A_b_, and A_c_ are the absorbances of the Pickering emulsion, blank, and control, respectively.

The Pickering emulsion was dissolved in ethanol at a ratio of 1:3 (*v*/*v*). Tris-HCl was then added, and the mixture was reacted for 20 min at 25 °C in a water bath. Next, pyrogallol was added to the mixture, and the reaction proceeded for 5 min. Finally, HCl was added, and the mixture was stirred well. The absorbance was measured at 299 nm. The O_2_^−^ radical scavenging activities (%) were calculated using the equation:$$O_{2}^{-}\;\left(\%\right)=\left(1-\frac{A_{s}-A_{c}}{A_{b}}\right)\times 100\%$$
where A_s_, A_b_, and A_c_ are the absorbances of the Pickering emulsion, blank, and control, respectively.

### 2.11. Statistical Analysis

The experiments and analyses were performed at least in triplicate, and the data were expressed as the mean ± standard deviation. The results were compared by one-way analysis of variance (ANOVA) with Duncan’s test using the Statistical Package for the Social Sciences 20.0 (SPSS Inc., Chicago, IL, USA). A value of *p* < 0.05 was considered statistically significant.

## 3. Results and Discussions

### 3.1. Mean Particle Size and ζ-Potential of the Pickering Emulsion

The mean particle size and ζ-potential of the Pickering emulsion after preparation was shown in previous research [21]. Figure 1 shows the changes in the mean particle size and ζ-potential of the Pickering emulsion under different environment stresses. In the pH stress test, the mean particle size of the Pickering emulsion was the smallest at pH 7 and the largest at pH 4 (Figure 1A). Similarly, the absolute ζ-potential was first increased and then decreased with the increasing of pH, in which the absolute ζ-potential was the highest at pH 7. These results possibly occurred because the complex particles remained intact under neutral conditions (pH 7) and were uniformly adsorbed on the surface of the oil droplets. However, at their isoelectric point (pH 4), the complex particles precipitated, aggregated, and desorbed from the oil droplet surfaces. In turn, the oil droplets aggregated, and the mean particle size of the Pickering emulsion increased. The mean particle sizes of the Pickering emulsion at pH 2 and pH 9 were higher than those at pH 7 and lower than those at pH 4. This indicated the protonation and deprotonation of the complex particles [26], which led to oil droplet aggregation and an increased mean particle size. However, at pH 2, the complex particle was not fully aggregated; therefore, the mean particle size was lower than that at pH 7 and pH 9.

The mean particle size of the Pickering emulsion at −20 °C was significantly larger than those at other temperatures (*p* < 0.05), as was the absolute ζ-potential (Figure 1B). No significant differences in the mean particle sizes at 25, 37, and 60 °C were observed. At low temperatures, the moisture in the Pickering emulsion and complex particles formed ice crystals of different sizes, which destroyed the structure of the composite particles and caused them to desorb from the oil droplet surfaces. Thus, the oil droplets aggregated, and the particle size increased. On the contrary, there was no significant change in the mean particle size at 60 °C, and the absolute value of the ζ-potential exceeded 30 mV, which indicated that the complex particles were less affected at high temperatures.

With increasing NaCl concentration, the mean particle size also increased, while the absolute ζ-potential decreased (Figure 1C). The addition of NaCl weakened the electrostatic repulsion between the droplets; hence, they were unable to overcome weak interactions, such as hydrogen bonds and van der Waals forces. When the NaCl concentration was increased to 500 mM, the mean particle size of the Pickering emulsion increased threefold compared to that of the untreated Pickering emulsion. The addition of saline ions to the emulsion resulted in a change in the surface charge of the emulsion. Therefore, aggregation occurred owing to the interaction between the hydrophobicity of the emulsion and the net charge of the emulsion droplet.

On comparing the effects of the three environmental stresses on the mean particle size and ζ-potential, we established that the mean particle size of the Pickering emulsion was <4 μm, except at −20 °C and 500 mM salt concentration. Moreover, the absolute ζ-potential under different temperature conditions were higher than those under different pH and salt concentration conditions. These results occurred because the pH weakened the steric hindrance to cause conformational contraction or carbohydrate depolymerization. On the contrary, the salt ions changed the electronegativity of the composite particles, which changed the potential of the emulsion.

### 3.2. Microstructure of the Pickering Emulsion

The microstructure of the Pickering emulsion was observed by CLSM and cryo-SEM (Figure 2 and Figure 3, respectively). The CLSM images were acquired by staining the oil phase red and the ChN–Fuc blue (Figure 2). At −20 °C, the image was mostly red with some blue scattered throughout (Figure 2(A1)). This result revealed that the oil was aggregated and the ChN–Fuc complex particles were destroyed. In Figure 2(A2,A3), blue is tightly wrapped around the red color, indicating that the complex particles completely encapsulated the oil droplets and stabilized the Pickering emulsion at 25 and 37 °C. Larger droplets are observed in Figure 2A and the droplet particle size distribution is uneven, indicating that some of the droplets aggregated but did not fuse. These results showed that the influence of low temperature on the emulsion is stronger than that of high temperature. This is because the ice crystals destroyed the o/w structure of the Pickering emulsion after the emulsion was frozen and thawed, resulting in leaching of the internal phase oil.

With increasing NaCl concentration, the particle size of the red-oil droplets also increased (Figure 2(B1–B4)). Here, the blue-complex particles of the Pickering emulsion broke away from the surface of the red-oil droplet at 50 and 100 mM. When the NaCl concentration was increased to 500 mM, numerous red droplets aggregated; however, the oil droplet surfaces were still covered with blue-complex particles. This result is consistent with that of the ζ-potential, wherein the ζ-potential of the Pickering emulsion at 500 mM was −10 mV. At the lower absolute value of the zeta-potential, the Pickering emulsion tended to aggregate or flocculate; hence, it was difficult for the complex particles to maintain the stability of the Pickering emulsion.

The Pickering emulsion (except at pH 7) displayed different degrees of aggregation; however, the blue-complex particles completely encapsulated the oil droplets, which did not leak. This was inconsistent with the particle size result because the aggregation of the individual emulsion particles caused the particle size to increase. The change in pH altered the surface charge of the Pickering emulsion, and the electrostatic repulsion between the droplets was reduced. This pH change did not cause the complex particles to fall off the surface of the oil droplets. Thus, the Pickering emulsion structure was broken, and the oil droplet aggregated at −20 °C. Under other environmental stress conditions, the Pickering emulsion maintained its complete structure, indicating that the composite particles also stabilized the emulsion under different environmental stresses.

The surface structure of the Pickering emulsion differed at different temperatures (Figure 3(A1–A4)). Thus, the surface of the Pickering emulsion at −20 °C was smooth, and freeze–thaw cycling led to particle redistribution on the o/w interface and desorption of the particles from the interface. This increased the probability of the droplets to coalesce in the system. A network structure was observed on the surface of the Pickering emulsion at 25, 37, and 60 °C; however, the network structure was aggregated at 60 °C. This phenomenon revealed that the Brownian motion of the complex particles was enhanced with the rise in temperature, which promoted the collision of the complex particles as well as aggregation [1]. The aggregation of the complex particles on the interface of the Pickering emulsion was also attributed to the strong hydrophobic interaction [27].

One of the environmental stresses normally encountered in food emulsions is the variable ionic strength. Thus, the interface structure of the Pickering emulsion was observed under different NaCl concentrations (Figure 3(B1–B4)). The oil droplet surfaces of the Pickering emulsion at 0 mM NaCl were encapsulated with a dense network structure. However, with increasing NaCl concentration, the network structure was destroyed, and large pores were observed because of the elevated hydrophobicity and electrostatic screening [27]. When the NaCl concentration was further increased to 500 mM, the complex particles accumulated on the surface of the oil droplets, and the network structure was completely destroyed. With the addition of salt ions, the complex particles on the surface of the Pickering emulsion charge were reduced by electrostatic screening. Furthermore, the complex particles aggregated when attractive interactions, such as van der Waals forces and hydrophobic effects (between the droplets), dominated the repulsion forces [28].

The microstructure of emulsions varies at different pH values (Figure 3(C1–C4)). There were particles attach to the surface of Pickering emulsion at pH 2, and the surface of Pickering emulsion was smooth at pH 4, 7, and 9. The environment around the droplet of the Pickering emulsion was different under the different pH value. There was aggregation around the Pickering emulsion at pH 2 and pH 9. However, the environment around Pickering emulsion at pH 7 was network. These results were attributed to the protonation of the amino groups on the ChN and F groups, which led to greater electrostatic repulsion between the polymer chains and the thinning of the interfacial film due to particle dissolution. This led to membrane rupture, which resulted in coalescence, oiling off, and eventual demulsification [27].

### 3.3. Chemical Stability of the Pickering Emulsion

By monitoring the POV and TBARS in the emulsion system, the ability of the ChN–Fuc composite particles to inhibit lipid oxidation was evaluated. With increasing storage time, the POV value of the Pickering emulsion gradually increased, while the TBARS value first increased and then decreased. This was attributed to the decrease in the rate of growth of the primary product, which caused a decrease in the content of final secondary product. The POV and TBARS values of the Pickering emulsion for the different pH values were lower than those for other environmental stresses (Figure 4A,D), indicating that the ChN–Fuc complex particles played a role in free radical scavenging and acted as chelating agents for metal ions, thereby inhibiting the oxidation of the oils [29]. After heating at 50 °C for several days, the oil was less susceptible to oxidation under acidic and alkaline conditions than under neutral conditions. The TBARS contents of the different emulsions exhibited different trends within 10 days. This was attributed to the fact that the emulsion at a neutral pH was more susceptible to oxidation, and the primary oxidation products were continuously converted into secondary oxidation products. Thus, the TBARS content reached a maximum on the fourth day. Subsequently, the secondary products were converted into other reducible components. On the contrary, the emulsions under acidic and alkaline conditions reached their maximum TBARS content at 6 d and 8 days, respectively. This was because the production rate of the primary oxidation products slowed down, which resulted in a decrease in the amount of secondary oxidation products.

The POV value of the Pickering emulsion at −20 °C was significantly higher than those at the other three temperatures (Figure 4B,E), which indicated the instability of the emulsion and the leakage of oil droplets at this temperature. The TBARS of the Pickering emulsion at −20 °C rose sharply between 0 and 2 d because of the continuous conversion of the POV to YBARS. The decrease in the TBARS content was due to the conversion of the secondary oxidation products into other substances. The trends of the POV at the other three temperatures were similar; however, the TBARS content was quite different from that at −20 °C. The POV formation at the other three temperatures was less than that at −20 °C for 0–2 days. The TBARS formation at the other three temperatures was less than that at −20 °C for 2–4 days. These results were attributed to the continuous transformation of the oxidation products into other substances; hence, their content was continuously reduced. In general, the 25 °C emulsion displayed superior oxidation stability, while the −20 °C emulsion displayed the lowest oxidation stability, which are observations that agree adequately with the above results.

The POV and TBARS contents of the emulsion were the lowest at 500 mM NaCl concentration (Figure 4C,F), which indicated that the emulsion effectively inhibited the oxidation of the oil droplets at this concentration. In contrast, the POV and TBARS contents of the emulsion at salt concentrations of 50 and 100 mM were the highest. These results were attributed to the aggregation of the complex particles on the surface of the Pickering emulsion at a concentration of 500 mM, which increased the thickness of the interface layer and prevented the oil droplets from contacting the external environment. However, at 50 and 100 mM, the interface-network structure of the emulsion was destroyed, which exposed the oil droplets to the external environment and hence increased the possibility of oxidization.

### 3.4. Physical Stability of the Pickering Emulsion

The physical stability of the Pickering emulsion under different environmental stresses was characterized by CI, physical stability, and storage stability (Figure 5). The CIs of the Pickering emulsion at pH 2 and pH 4 were higher than those at pH 7 and pH 9. Similarly, at pH 2 and pH 4, the physical stability was lower, the particle size was larger, and more water was separated out after 30 days compared to the results for pH 7 and pH 9 (Figure 5A,D,G). These results demonstrated that the stability of the Pickering emulsion was superior for alkaline conditions compared to that for acidic conditions. The sulfate group in ChN–Fuc was ionized under neutral and alkaline conditions, and the complex particles repelled each other to generate greater steric hindrance. Moreover, with increasing pH, the electrostatic interaction between the charges was enhanced, and the repulsive force was transferred from the positive charge (lower pH) to the negative charge (higher pH), resulting in the restabilization of the emulsion.

At temperatures of 25, 37, and 60 °C, the physical stability of the emulsion was ~65%, which was 20% higher than that observed at −20 °C (Figure 5E). This was because the emulsion droplet structure was destroyed after freezing/thawing and did not resist the centrifugal force. Moreover, the Pickering emulsion exhibited an appreciable droplet dispersion at these three temperatures, and the repulsive force between the droplets stabilized the emulsion. In contrast, the Pickering emulsion at −20 °C exhibited significant stratification on the first day, and the CI approached 50%. After 7 days, the CI approximated 55%, indicating that the emulsion was extremely unstable. Conversely, at the other three temperatures, the CI of the Pickering emulsion was approximately 3% for 1 days and only slightly increased to <10% for 7 days. This small change was related to the small average particle size of the emulsion droplets and appropriate ζ-potential, which maintained high emulsion stability. Figure 5H illustrates that the emulsion stored at −20 °C was stratified, and the water was separated out. Here, the sample was no longer a conventional emulsion, and the measured average particle size decreased to <1 μm. In contrast, the emulsions stored at the other three temperatures remained stable after 30 d without delamination, and the average particle size changed little, indicating that these three emulsions exhibited excellent storage stability.

The CI value of the Pickering emulsion at 500 mM NaCl was significantly higher than those at the other salt concentrations. With increasing storage time, the CI value of the emulsion with 100 mM NaCl did not change significantly, while that with 500 mM salt gradually increased. The change in the physical stability of the Pickering emulsion at low salt concentration was <10%, while that at 500 mM NaCl was significant (30% decrease). The results demonstrated that at low salt concentrations, the Pickering emulsion was stabilized by the complex particles through electrostatic repulsion [30]. The mean particle size of the Pickering emulsion at salt concentrations of 0–100 mM did not change after storage for 30 days. However, the mean particle size at 500 mM NaCl increased by 2 μm after 30 days, owing to the aggregation of the complex particles on the droplet surfaces at high salt concentration, which caused the particle size to increase [31].

### 3.5. Antioxidant Activities of the Pickering Emulsion

The free radical scavenging rates of ABTS^+^, DPPH, and O_2_^−^ were used to evaluate the antioxidant activity of the Pickering emulsion under different environmental stresses (Figure 6). The free radical scavenging rates of the Pickering emulsion under the above-described environment stresses increased with increasing concentration. The free radical scavenging rate of the Pickering emulsion at pH 7 was higher than those of the other three pH values, indicating that Fuc was not destroyed at pH 7 and still exhibited antioxidant activity. The free radical scavenging rates of the Pickering emulsion under different salt concentrations decreased with increasing salt concentration, thereby destroying the interface-network structure of the oil droplets. Moreover, the complex particles aggregated and fell off the oil droplet surface, which not only destroyed the stability of the Pickering emulsion but also reduced its antioxidant activity. The free radical scavenging rate of the Pickering emulsion at −20 °C was lower than those at the other temperatures. This was because the Pickering emulsion structure was broken at −20 °C and the complex particles were mixed with oil and water, thereby reducing the contact area with ABTS^+^, DPPH, and O_2_^−^, which reduced the ratio of scavenging free radicals. In summary, the antioxidant activity of Pickering emulsion was the highest under pH 7 and 25°C. Moreover, the ABTS^+^ scavenging rate of the Pickering emulsion was the highest under the 0 mM salt concentration, and the DPPH and O_2_^−^ scavenging rate of the Pickering emulsion was the highest under 100 mM salt concentration. Cao et al. also reported that the Pickering emulsion with selenium nanoparticles exhibited excellent antioxidant properties [32]. Fuc mainly activate the antioxidant enzyme system, increase the activity of antioxidant enzymes, and reduce oxidation products. Changes in environmental conditions lead to differences in the molecular weight of Fuc as well as the content of sulfates, thus resulting in differences in the antioxidant activity of the Pickering emulsions. Therefore, the Fuc-ChN Pickering emulsion with excellent antioxidant activity can be used in cosmetics and food industries.

## 4. Conclusions

A Pickering emulsion with two oppositely charged composite particles (ChN and F) as emulsifiers was fabricated and its basic properties, microstructure, stabilities (physical and chemical), and antioxidant activity were investigated under different environmental stresses. The pH significantly affected the stability and antioxidant activity of the emulsion. Under neutral conditions, the emulsion displayed the smallest particle size, largest electronegativity, and uniform droplet distribution. Under acid/alkaline conditions, the structure of the ChN–Fuc composite particles was destroyed, which affected the stability and antioxidant activity of the emulsion. In general, the emulsion stability and antioxidant activity were in the order of neutral > alkaline > acid > isoelectric point. The temperature endowed the emulsion stability and antioxidant activity with a polarizing trend. After freezing/thawing at −20 °C, the emulsion was unstable, and the physical and chemical stabilities and antioxidant activity of the emulsion were low. In contrast, the physical and chemical stabilities and antioxidant activity of the emulsion at 25, 37, and 60 °C were superior with minor difference in between them. With increasing salt ion concentration, the particle size of the emulsion increased, while the ζ-potential decreased. Moreover, the physical stability and antioxidant activity of the emulsion gradually decreased; however, at a high salt concentration of 500 mM, a superior oxidation stability of the Pickering emulsion was obtained. Oppositely charged polysaccharides can be used as stabilizers for carrying Pickering emulsions, providing a theoretical basis for the fabrication of new Pickering emulsions.

## Figures and Tables

**Figure 1 foods-11-01835-f001:**
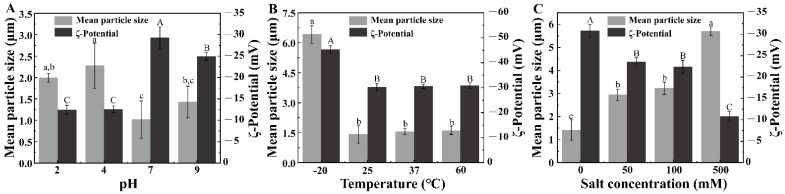
Mean particle size and ζ-potential of the Pickering emulsion under different environmental stresses. (**A**) pH, (**B**) temperature, and (**C**) salt concentration. Different letters indicate significant differences (*p* < 0.05).

**Figure 2 foods-11-01835-f002:**
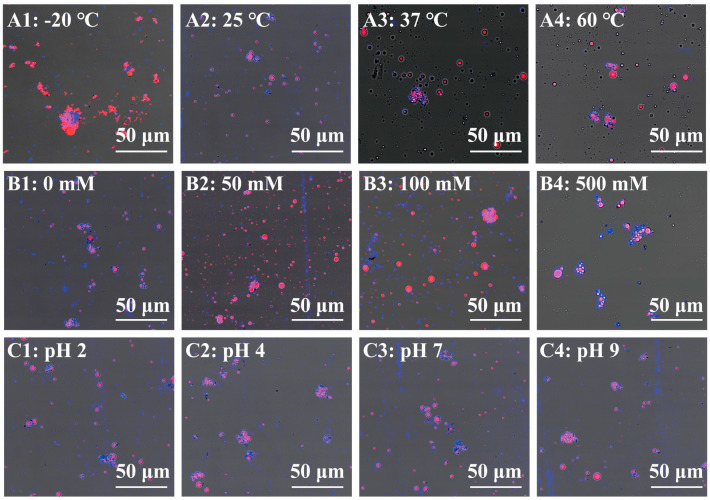
Confocal laser scanning microscopy (CLSM) images of the Pickering emulsion under different environmental stresses. (**A1**–**A4**) pH, (**B1**–**B4**) temperature, and (**C1**–**C4**) salt concentration.

**Figure 3 foods-11-01835-f003:**
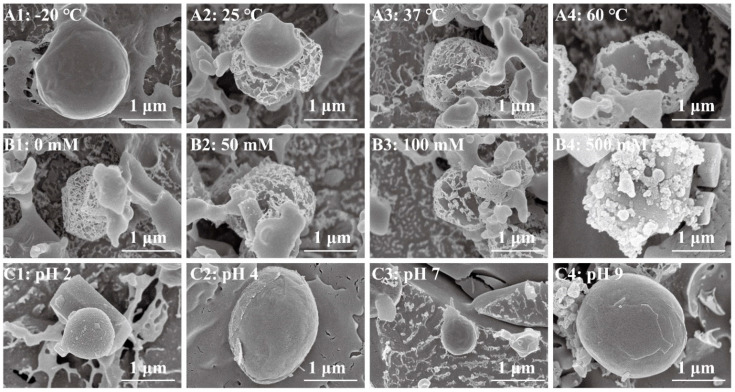
Cryo-scanning electron microscopy (Cryo-SEM) images of the Pickering emulsion under different environmental conditions. (**A1**–**A4**) temperature, (**B1**–**B4**) salt concentration, and (**C1**–**C4**) pH.

**Figure 4 foods-11-01835-f004:**
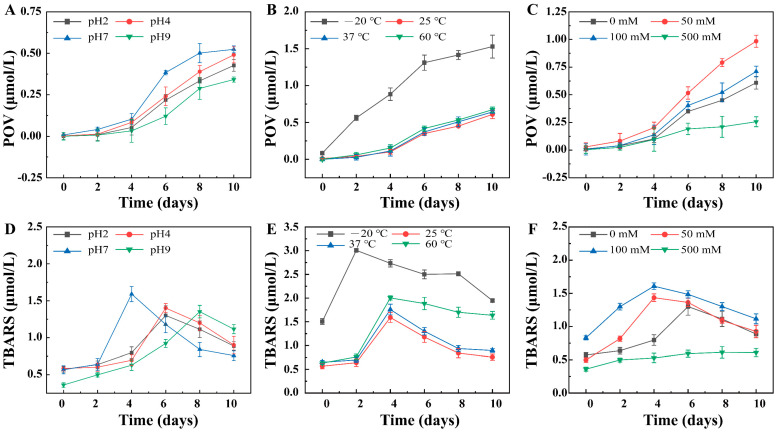
Chemical stability of the Pickering emulsion under different environmental stresses. (**A**–**C**) Peroxide value (POV) and (**D**–**F**) thiobarbituric acid reactive substance (TBARS) of the Pickering emulsion under different pH, temperature, and salt concentration conditions.

**Figure 5 foods-11-01835-f005:**
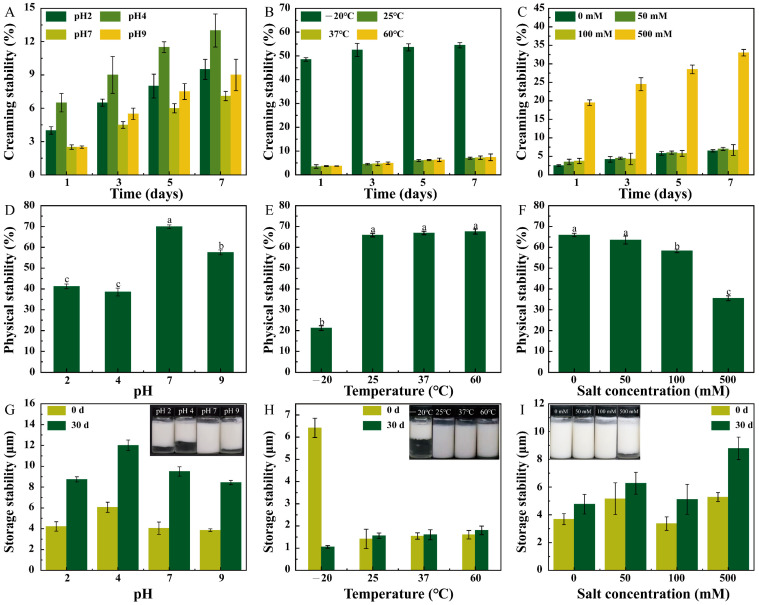
Physical stability of the Pickering emulsion under different environmental stresses. (**A**–**C**) creaming index (CI), (**D**–**F**) physical stability, and (**G**–**I**) storage stability of the Pickering emulsion under different pH, temperature, and salt concentration conditions. Different letters indicate significant differences (*p* < 0.05).

**Figure 6 foods-11-01835-f006:**
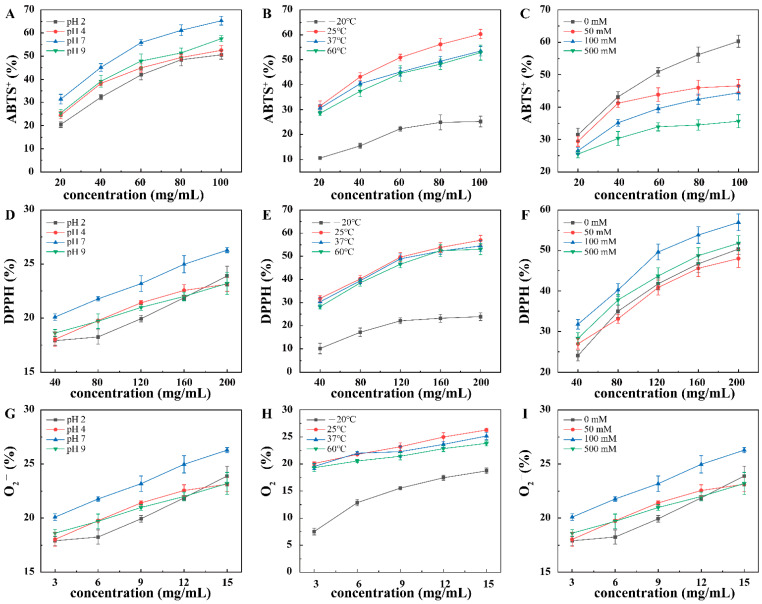
Antioxidant activities of the Pickering emulsion under different environmental stresses. (**A**–**C**) 2,2′-Azino-bis(3-ethylbenzothiazoline-6-sulfonate) (ABTS^+^) cations, (**D**–**F**) 2,2-diphenyl-1-picrylhydrazyl (DPPH), and (**G**–**I**) O_2_^−^ anions of the Pickering emulsion under different pH, temperature, and salt concentration conditions.

## Data Availability

The datasets generated for this study are available on request to the corresponding author.

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
