# Peer review of "Oppositely Charged Pickering Emulsion Co-Stabilized by Chitin Nanoparticles and Fucoidan: Influence of Environmental Stresses on Stability and Antioxidant Activity"

_foods, 2022, doi:10.3390/foods11131835_

Round 1
Reviewer 1 Report
1. What is the main question examined in the research?
Oppositely charged Pickering emulsion co-stabilized by chitin nanoparticles and fucoidan: Influence of environmental stresses on stability and antioxidant activity
2. Do you consider the subject to be original or related to the field, and if so, why?
Yes, according to the article review, this is a topic related to the field of food industry and there are few articles with this precision about Pickering emulsion.
3. What adds to the subject area compared to other published content?
The production of nanoemulsions and the study of oxidative properties are very important in the science of food industry, and this article illustrates this well.
4. In line 209, how much did the mean particle size differ?
5. Figure 1 is not of good quality
Conclusions and sources are well described.
As a result, Pickering emulsion with two composite particles with opposite charge (ChN and F) and its antioxidant properties have been investigated and described correctly.
But the volume of content is very large and it is necessary to omit additional explanations in some places.
Author Response
- What is the main question examined in the research?
Response: We thank the reviewer for the question. To overcome the limitations of conventional emulsions, we previously fabricated a Pickering emulsion from positively charged ChNs and negatively charged fucoidan. We verified that the as-synthesized Pickering emulsion comprising oppositely charged particles possessed excellent physical and storage stabilities. In the present study, we proceeded to investigate the influence of environmental stresses on the stability of this Pickering emulsion and predicted that the Pickering emulsion would exhibit significant resistance under different environmental stresses, such as pH, heat, and presence of salts. It is necessary to provide theoretical basis for industrial production
- Do you consider the subject to be original or related to the field, and if so, why?
Yes, according to the article review, this is a topic related to the field of food industry and there are few articles with this precision about Pickering emulsion.
- What adds to the subject area compared to other published content?
The production of nanoemulsions and the study of oxidative properties are very important in the science of food industry, and this article illustrates this well.
- In line 209, how much did the mean particle size differ?
Response: We thank the reviewer for the question. The mean particle size was increased from 1.41 μm to 5.7 μm with the increasing of salt concentration.
- Figure 1 is not of good quality
Response: We thank the reviewer for the question. We have modified the Fig. 1 in the revised manuscript.
Conclusions and sources are well described.
As a result, Pickering emulsion with two composite particles with opposite charge (ChN and F) and its antioxidant properties have been investigated and described correctly.

Reviewer 2 Report
A brief summary
The article „Oppositely charged Pickering emulsion co-stabilized by chitin nanoparticles and fucoidan: Influence of environmental stresses on stability and antioxidant activity“ is focused on the description of o/w emulsions (stabilized by chitin nanoparticles and fucoidan) behaviour under stresses – pH, temperature, salt.
General concept comments
The idea to evaluate the influence of common stresses which can ocur during manufacturing is interesting and appropriate for section Food Physics and (Bio)Chemistry in Foods. But I suggest some revision for better interpretation of measured results.
Specific comments:
Abstract
Line 13: I recommend to specify a type of emulsion – o/w
Introduction
line 61: I suggest to mention briefly the success rate of reported examples
line 67: Clarify „conventional emulsions“
line 90: Here, I also recommend to specify a type of emulsion – o/w
Materials and Methods
line 105: Which device was used for zeta-potential measurement - Zetasizer? Mastersizer does not allow zeta-potential measurement …
Results and discussion
line 186: I suggest to supply that these data were measured after preparation
line 200: I am afraid the data were not collected correctly … especially zeta-potential of frozen samples – Can you admit the possibility of zeta-potential measurement in frozen samples by any citation?
line 240: w/o?
line 248: at lower or at lower absolute value of zeta-potential?
line 299: Fig.3: Cryo instead of Cyro
line 349: Authors mention in 2.9 zeta potential measurement during storage – but this measurement is not discussed
line 391: Fig.5: Generally, I suggest better explanation in „Methods“ of data collection during storage – results were measured after 1, 3 ….days
Conclusions
I recommend to highlight the main benefits of this current work
Author Response
A brief summary
The article “Oppositely charged Pickering emulsion co-stabilized by chitin nanoparticles and fucoidan: Influence of environmental stresses on stability and antioxidant activity” is focused on the description of o/w emulsions (stabilized by chitin nanoparticles and fucoidan) behaviour under stresses – pH, temperature, salt.
General concept comments
The idea to evaluate the influence of common stresses which can occur during manufacturing is interesting and appropriate for section Food Physics and (Bio)Chemistry in Foods. But I suggest some revision for better interpretation of measured results.
Specific comments:
Abstract
Line 13: I recommend to specify a type of emulsion – o/w
Response: We thank the reviewer for the comments. we have added the specify emulsion type in the revised manuscript as follows:
Oppositely charged chitin nanoparticles and fucoidan complex particles were used as emulsifiers to stabilize a O/W Pickering emulsion and explore its stability and antioxidant activity under different environmental stresses.
Introduction
line 61: I suggest to mention briefly the success rate of reported examples
Response: We thank the reviewer for the comments. We have modified the sentences in the revised manuscript as follows:
Meanwhile, related research has been successfully fabricated the electrostatic complexation relevant to food include mixtures of proteins and strong polyanions [17], whey proteins and exocellular polysaccharide EPS B40 [18], chitosan-alginate [19], and xan-than gum and whey proteins [20].
line 67: Clarify “conventional emulsions”
Response: We thank the reviewer for the comments. Conventional emulsions contain lipid droplets with mean droplet diameters greater than about 200 nm.
References:
Paul, D., Dey, T. K., Mukherjee, S., Ghosh, M., & Dhar, P. (2014). Comparative prophylactic effects of α-eleostearic acid rich nano and conventional emulsions in induced diabetic rats. Journal of food science and technology, 51(9), 1724-1736.
line 90: Here, I also recommend to specify a type of emulsion – o/w
Response: We thank the reviewer for the comments. we have added the specify emulsion type in the revised manuscript as follows:
The coarse emulsion was then further homogenized at 100 MPa for one cycle (FPG12805, Standard Fluid Power Ltd., England, UK) to form the O/W Pickering emulsion.
Materials and Methods
line 105: Which device was used for zeta-potential measurement - Zetasizer? Mastersizer does not allow zeta-potential measurement …
Response: We thank the reviewer for the mistake. We have revised the comments in the manuscript as follows:
The particle size and ζ-potential of the Pickering emulsion were determined using dynamic light scattering (Zetasizer Nano Zs, Malvern Instruments Ltd., Malvern, Worcestershire, UK).
Results and discussion
line 186: I suggest to supply that these data were measured after preparation
Response: We thank the reviewer for the comments. The mean particle size and zeta potential of Pickering emulsion after preparation was published in the Journal of the Science of Food and Agriculture title “Oil-in-water Pickering emulsion stabilization with oppositely charged polysaccharide particles: chitin nanocrystals/fucoidan complexes”. The published article was cited in the manuscript.
line 200: I am afraid the data were not collected correctly … especially zeta-potential of frozen samples – Can you admit the possibility of zeta-potential measurement in frozen samples by any citation?
Response: We thank the reviewer for the question. Zeta potential value measured at 25℃ after the sample has been frozen. Meanwhile, Aoki et al also measured the stability at -20℃.
References:
Aoki, T., Decker, E. A., & McClements, D. J. (2005). Influence of environmental stresses on stability of O/W emulsions containing droplets stabilized by multilayered membranes produced by a layer-by-layer electrostatic deposition technique. Food Hydrocolloids, 19(2), 209-220.
line 240: w/o?
Response: Thank the reviewer for pointing the mistake. We have revised the comments in the manuscript as follows:
This is because the ice crystals destroyed the O/W structure of the Pickering emulsion after the emulsion was frozen and thawed, resulting in leaching of the internal phase oil.
line 248: at lower or at lower absolute value of zeta-potential?
Response: Thank the reviewer for the comments. We have revised the comments in the manuscript as follows:
At lower absolute value of zeta-potential, the Pickering emulsion tended to aggregate or flocculate; hence, it was difficult for the complex particles to maintain the stability of the Pickering emulsion.
line 299: Fig.3: Cryo instead of Cyro
Response: Thank the reviewer for the comments. We have revised the comments in the manuscript as follows:
Fig. 3 Cryo -scanning electron microscopy (Cryo-SEM) images of the Pickering emulsion under different environ-mental conditions. (A1–A4) pH, (B1–B4) temperature, and (C1–C4) salt concentration.
line 349: Authors mention in 2.9 zeta potential measurement during storage – but this measurement is not discussed
Response: Thank the reviewer for pointing the mistake. Mean particle size was used to characterize the storage stability of Pickering emulsion, but the zeta potential measurement during storage in section 2.9 due to our careless. Therefore, we revised the methods of section 2.9 in the manuscript as follows:
The fresh Pickering emulsion was stored at 25 ℃ for 30 d, and its particle size were measured according to the methods described in section 2.3 to determine the storage stability.
line 391: Fig.5: Generally, I suggest better explanation in “Methods” of data collection during storage – results were measured after 1, 3 ….days
Response: Thank the reviewer for the comments. We have revised the methods in the manuscript as follows:
The fresh Pickering emulsion was placed in a colorimetric tube and sealed with a glass stopper and incubated at 25 ℃ for 1, 3, 5, and 7 d.
Conclusions
I recommend to highlight the main benefits of this current work
Response: Thank the reviewer for the comments. We have revised the methods in the manuscript as follows:
This result provides a theoretical basis for the application of Pickering emulsions in industrial production

Reviewer 3 Report
The authors perform an extensive experimental study on the Pickering emulsion generated by chitin nanocrystals/fucoidan complexes
and oils. The authors investigate the stability function of different environmental stresses. This work is a continuation of the work already published at reference 21. The manuscript is not ready for publication as is. I think it needs serious improvement. The manuscript is written in such a way that the author is lost in a plethora of details, but it doesn't capture the essence. Please see my observations below:
lines 200-201: "However, at pH 4, the complex particle was not fully aggregated, therefore, the 200
mean particle size was lower than that at the isoelectric point" I think there is a mistake< perhaps the authors meant pH 2 instead of pH 4.
In Figure 1A, the error bars for the mean particle size at pH 4 is so large that it overlaps with the mean value at pH 2, so I think the
conclusions drawn, that at pH2 the mean particle size is smaller than that at pH4 is not valid. THe authors should show in SI the graph with
the particle size distribution, so the reader has access to a more honest representation of data.
It is not clear to me, how the experiments at lines 202-209 were carried. Was the sample first heated and cooled and then the DLS experiment
was performed, or DLS experiment were carried at the temperatures given?
The statemetn at lines 216-218:"In this case, the neutralization re- 216
action between the negatively charged complex particles and salt ions reduced both the 217
amount of adsorbed composite particles on the oil droplets and the absolute value of the 218
emulsion ζ-potential" is really confusing. There is no "neutralization" reaction between the NaCl ions and the charged particles. This statement
is false, any undergraduate chemistry student will know that.
Also, when performing DLS and zeta potential measurements on an emulsion in different conditions, most of the results are going to be a convolution
between that of particles and those of droplets. It is clear that the authors intend to follow the results from the emulsion droplets, but
it is not clear how they have isolated the contribution to the zeta potential value of the non-adsorbed, freely moving particles in solution.
The authors should clear this aspect.
Figure 3. C1-C4. is not referenced in the text. Also wrong labels in the Figure 3C1-C4, instead of B2-B4.
Actually, the discussion at lines 296-299 is rather meaningless, what the reader observes in the SEM images does not correspond with what is
written/claimed in the text, for example: "At pH 9, the network-like composite particles were de- 296
stroyed; subsequently, they accumulated on the droplet surfaces in a chain shape. Gener- 297
ally, the Pickering emulsion droplets exhibited superior interface structure in neutral and 298
alkaline aqueous media. 29", where are these chain shapes accumulated on the droplet surface?
In section 3.5. the antioxidant activity of the Pickering emulsions is not well presented.
For example the authors, don't mention nor discuss, due to what kind of phenomenon, or mechanism are the obtained Pickering emulsions
expected to have antioxidant activity. Honestly, this one key point of the article is not properly discussed and presented.
This draft lacks the quality of a scientific work and cannot be published as is. The conclusions are not sound, and this is a mere presentation
of a series of experiments, as in a labbook.
Author Response
Reviewer 1
The authors perform an extensive experimental study on the Pickering emulsion generated by chitin nanocrystals/fucoidan complexes and oils. The authors investigate the stability function of different environmental stresses. This work is a continuation of the work already published at reference 21. The manuscript is not ready for publication as is. I think it needs serious improvement. The manuscript is written in such a way that the author is lost in a plethora of details, but it doesn't capture the essence. Please see my observations below:
lines 200-201: "However, at pH 4, the complex particle was not fully aggregated, therefore, the 200 mean particle size was lower than that at the isoelectric point" I think there is a mistake< perhaps the authors meant pH 2 instead of pH 4.
Response: Thank the reviewer for the comments. We have modified the sentence in the revised manuscript as follows:
However, at pH 2, the complex particle was not fully aggregated, therefore, the mean particle size was lower than that at pH 7 and pH 9.
In Figure 1A, the error bars for the mean particle size at pH 4 is so large that it overlaps with the mean value at pH 2, so I think the conclusions drawn, that at pH2 the mean particle size is smaller than that at pH4 is not valid. The authors should show in SI the graph with the particle size distribution, so the reader has access to a more honest representation of data.
Response: Thank to the reviewer for the comments. We agree with the reviewer’s comments. there was no significant difference in statistics. Therefore, the mean particle size was not compared at pH 2 and pH 4.
It is not clear to me, how the experiments at lines 202-209 were carried. Was the sample first heated and cooled and then the DLS experiment was performed, or DLS experiment were carried at the temperatures given?
Response: We thank the reviewer for the question. The sample first heated and cooled and then the DLS experiment was performed.
The statemetn at lines 216-218:"In this case, the neutralization re- 216 action between the negatively charged complex particles and salt ions reduced both the 217 amount of adsorbed composite particles on the oil droplets and the absolute value of the 218 emulsion ζ-potential" is really confusing. There is no "neutralization" reaction between the NaCl ions and the charged particles. This statement is false, any undergraduate chemistry student will know that.
Response: we thank the reviewer for the comments. We were also aware of this mistake and modified the sentences in revised manuscript as follows:
The addition of saline ions to the emulsion resulted in a change in the surface charge of the emulsion. Therefore, polymerization occurred owing to the interaction between the hydrophobicity of the emulsion and the net charge of the emulsion droplet.
Also, when performing DLS and zeta potential measurements on an emulsion in different conditions, most of the results are going to be a convolution between that of particles and those of droplets. It is clear that the authors intend to follow the results from the emulsion droplets, but it is not clear how they have isolated the contribution to the zeta potential value of the non-adsorbed, freely moving particles in solution. The authors should clear this aspect.
Response: Thanks for the comments of the reviewers, we strongly agree with the reviewer's suggestion. Emulsion is a mixture system including non-adsorbed, freely moving particles. Therefore, we will further separate the particles in the emulsion and explore the effect of each particle on the stability of the emulsion.
Figure 3. C1-C4. is not referenced in the text. Also wrong labels in the Figure 3C1-C4, instead of B2-B4.
Response: Thank to the reviewer for the question. The Fig. 3 had been modified in the revised manuscript.
Actually, the discussion at lines 296-299 is rather meaningless, what the reader observes in the SEM images does not correspond with what is written/claimed in the text, for example: "At pH 9, the network-like composite particles were de- 296 stroyed; subsequently, they accumulated on the droplet surfaces in a chain shape. Gener- 297 ally, the Pickering emulsion droplets exhibited superior interface structure in neutral and 298 alkaline aqueous media. 29", where are these chain shapes accumulated on the droplet surface?
Response: Thank to the reviewer for the comments. According to the comments, we modified the sentences in the revised manuscript as follows:
At pH 9, there was large particle around the droplet which attributed that the complex was aggregated.
In section 3.5. the antioxidant activity of the Pickering emulsions is not well presented. For example the authors, don't mention nor discuss, due to what kind of phenomenon, or mechanism are the obtained Pickering emulsions expected to have antioxidant activity. Honestly, this one key point of the article is not properly discussed and presented.
Response: Thank to the reviewer for the comments. according to the comments, the antioxidant activity of the Pickering emulsion was represented in the revised manuscript as follows:
The free radical scavenging rates of ABTS+, DPPH, and O2− were used to evaluate the antioxidant activity of the Pickering emulsion under different environmental stresses (Fig. 6). The free radical scavenging rates of the Pickering emulsion under the above-described environment stresses increased with increasing concentration. The free radical scavenging rate of the Pickering emulsion at pH 7 was higher than those of the other three pH values, indicating that Fuc was not destroyed at pH 7 and still exhibited antioxidant activity. The free radical scavenging rates of the Pickering emulsion under different salt concentrations decreased with increasing salt concentration, thereby de-stroying the interface-network structure of the oil droplets. Moreover, the complex particles aggregated and fell off the oil droplet surface, which not only destroyed the stability of the Pickering emulsion but also reduced its antioxidant activity. The free radical scavenging rate of the Pickering emulsion at −20 ℃ was lower than those at the other temperatures. This was because the Pickering emulsion structure was broken at −20 °C and the complex particles were mixed with oil and water, thereby reducing the contact area with ABTS+, DPPH, and O2−, which reduced the ratio of scavenging free radicals. Fuc mainly activate the antioxidant enzyme system, increase the activity of antioxidant enzymes, and reduce oxidation products. Changes in environmental con-ditions lead to differences in the molecular weight of Fuc as well as the content of sul-fates, thus resulting in differences in the antioxidant activity of the Pickering emulsions.
This draft lacks the quality of a scientific work and cannot be published as is. The conclusions are not sound, and this is a mere presentation of a series of experiments, as in a lab book.
Response: Thank the reviewer for these comments and question. We modified the contents to improve the quality of manuscript.

Reviewer 4 Report
Referee report
Title: Oppositely charged Pickering emulsion co-stabilized by chitin 2 nanoparticles and fucoidan: Influence of environmental 3 stresses on stability and antioxidant activity
Dear Editor,
This paper presents interesting results which can be useful for different kind of food industry but the way of presentation should be better. This paper can be recommended to publish after minor revision and in my opinion the following changes should be done before the paper acceptance.
Abstract
Instead of O/W should be o/w
Line 60-64
Literature is too old from 1996-2000
Some new literature items should be added. Please refer in the introductory part and in the discussion the analogous results. For example:
Effect of ionic strength on electrokinetic properties of oil/water emulsions with dipalmitoylphosphatidylcholine, A: Physicochemical and Engineering Aspects Colloids and Surfaces 302 (2007) 141-149
Effect of temperature on n-tetradecane emulsion in the presence of phospholipid DPPC and enzyme lipase or phospholipase A2, Langmuir 24(14) (2008) 7413-7420
Electrokinetic properties of n-tetradecane/ethanol emulsions with DPPC and enzyme Lipase or Phospholipase A2, A: Physicochemical and Engineering Aspects Colloids and Surfaces 332 (2009) 150-156
Comparison of n-Tetradecane/Electrolyte Emulsions Properties Stabilized by DPPC and DPPC vesicles in the electrolyte solution” Colloids and Surfaces B: Biointerfaces 83 (2011) 108-115
Influence of Dipalmitoylphosphatidylcholine (or Dioleolyphosphatidylcholine) and Phospholipase A2 Enzyme on the Properties of Emulsions” Journal of Colloid Interface Science 373 (2012) 75-83
Edible films made from blends of gelatin and polysaccharide-based emulsifiers - A comparative study, Food Hydrocolloids 96 (2019) 555-567
Effect of chitosan, hyaluronic acid and/or titanium dioxide on the physicochemical characteristic of phospholipid film/glass surface, Physicochem. Probl. Miner. Process. 55(6) (2019) 1535–1548
Materials and methods
pH2
Specify how this pH was obtained, what amount of acid, does it interfere with food tests (such a low pH)
Line 108
Explain why was such a value of the refractive index (1.46) chosen, which particles were decisive here
Line 145, 159, 166, 174
Please specify where these different values at which the absorbance was determined. For each of them, explain your choice, for which compound there is a specific maximum peak, or what other reason is there?
Line 158, Line 164
ABTS, DPPH
Recall these shortcuts again
Line 191
This sentence is too general. The changes to these 2 parameters are completely different. From a mathematical point of view, how can we compare diameters that can only take positive values with a zeta potential that can take positive, negative, or 0 values. This is illogical and incompatible with the foundations of mathematics. I understand that colloquial wording has crept in here. Expand your thought more scientifically.
Fig. 1
These extra letters on the bars in the chart are confusing and have not been explained in the body text.
Line 239
Please standardize the numbering of figures throughout the text.
Sentence should be singular, English should be carefully corrected along all manuscript.
Line 304
Recall these shortcuts
Line 320
Explain
Line 397, 411
Index should be
Line 402
Precise?
Line 434
Is too general. Please emphasize in the Conclusions section how this paper contributes to new fundamental understanding for food domain.
Bibliography should be more improved
I can recommend this article, but after minor revision.

Author Response
Reviewer 2
Referee report
Title: Oppositely charged Pickering emulsion co-stabilized by chitin 2 nanoparticles and fucoidan: Influence of environmental 3 stresses on stability and antioxidant activity
Dear Editor,
This paper presents interesting results which can be useful for different kind of food industry but the way of presentation should be better. This paper can be recommended to publish after minor revision and in my opinion the following changes should be done before the paper acceptance.
Abstract
Instead of O/W should be o/w
Response: We thank the reviewer for the comments. we have modified the O/W to o/w.
Line 60-64
Literature is too old from 1996-2000
Some new literature items should be added. Please refer in the introductory part and in the discussion the analogous results. For example:
Effect of ionic strength on electrokinetic properties of oil/water emulsions with dipalmitoyl phosphatidylcholine, A: Physicochemical and Engineering Aspects Colloids and Surfaces 302 (2007) 141-149
Effect of temperature on n-tetradecane emulsion in the presence of phospholipid DPPC and enzyme lipase or phospholipase A2, Langmuir 24(14) (2008) 7413-7420
Electrokinetic properties of n-tetradecane/ethanol emulsions with DPPC and enzyme Lipase or Phospholipase A2, A: Physicochemical and Engineering Aspects Colloids and Surfaces 332 (2009) 150-156
Comparison of n-Tetradecane/Electrolyte Emulsions Properties Stabilized by DPPC and DPPC vesicles in the electrolyte solution” Colloids and Surfaces B: Biointerfaces 83 (2011) 108-115
Influence of Dipalmitoylphosphatidylcholine (or Dioleolyphosphatidylcholine) and Phospholipase A2 Enzyme on the Properties of Emulsions” Journal of Colloid Interface Science 373 (2012) 75-83
Edible films made from blends of gelatin and polysaccharide-based emulsifiers - A comparative study, Food Hydrocolloids 96 (2019) 555-567
Effect of chitosan, hyaluronic acid and/or titanium dioxide on the physicochemical characteristic of phospholipid film/glass surface, Physicochem. Probl. Miner. Process. 55(6) (2019) 1535–1548
Response: Thank the reviewer for the comments. According to the reviewer’s comment, the reference has been replaced.
Materials and methods
pH2
Specify how this pH was obtained, what amount of acid, does it interfere with food tests (such a low pH)
Response: the pH was adjusted using HCl. And the ChN-Fuc complex was prepared at pH 2 and then freeze-drying. But the Pickering emulsion was prepared at pH 7. The condition did not interfere with food tests.
Line 108
Explain why was such a value of the refractive index (1.46) chosen, which particles were decisive here
Response: the refractive index was determined according to the methods of Xiong et al.
Reference:
Xiong, Y., Li, Q., Miao, S., Zhang, Y., Zheng, B., & Zhang, L. (2019). Effect of ultrasound on physicochemical properties of emulsion stabilized by fish myofibrillar protein and xanthan gum. Innovative Food Science & Emerging Technologies, 54, 225-234.
Line 145, 159, 166, 174
Please specify where these different values at which the absorbance was determined. For each of them, explain your choice, for which compound there is a specific maximum peak, or what other reason is there?
Response: Thank to the reviewer for the question. ABTS is oxidized to green ABTS+ under the action of an appropriate oxidant. The total antioxidant capacity of the sample can be determined and calculated by measuring the absorbance of ABTS+ at 734nm.
As a stable free radical, DPPH can exist stably in organic solvents. Its alcoholic solution is purple, and it needs to be stored in the dark at low temperature. It has a single electron, so it can accept one electron or hydrogen ion. It has a wavelength of 517nm. maximum absorption.
O2- reacts with hydroxylamine to generate NO2, NO2 generates pink azo substances under the action of p-aminobenzenesulfonic acid and aniline, and the absorbance value is measured at 530 nm.
Line 158, Line 164
ABTS, DPPH
Recall these shortcuts again
Response: Thank to the reviewer for the comments. The ABTS and DPPH was revised.
Line 191
This sentence is too general. The changes to these 2 parameters are completely different. From a mathematical point of view, how can we compare diameters that can only take positive values with a zeta potential that can take positive, negative, or 0 values. This is illogical and incompatible with the foundations of mathematics. I understand that colloquial wording has crept in here. Expand your thought more scientifically.
Response: Than to the reviewer for the comments. We modified the sentences in the revised manuscript as follows:
Similarly, the absolute ζ-potential was first increased and then decreased with the in-creasing of pH, in which the absolute ζ-potential was the highest at pH 7.
Fig. 1
These extra letters on the bars in the chart are confusing and have not been explained in the body text.
Response: We thank the reviewer for the comments. Means with different superscript letters within the same column differ significantly.
Line 239
Please standardize the numbering of figures throughout the text.
Sentence should be singular, English should be carefully corrected along all manuscript.
Response: Thank to the reviewer for the comments. We have revised the manuscript.
Line 304
Recall these shortcuts
Response: Thank to the reviewer for the comments. We have revised the manuscript.
Line 320 Explain; Line 397, 411; Index should be; Line 402 Precise? ; Line 434
Response: Thank to the reviewer for the comments. We have revised the manuscript according to the reviewer’s comments.
Is too general. Please emphasize in the Conclusions section how this paper contributes to new fundamental understanding for food domain.
Bibliography should be more improved
Response: Thank to the reviewer for the comments. According to the comments, the conclusion section was modified in the revised manuscript as follows:
Oppositely charged polysaccharides can be used as stabilizers for carrying Pickering emulsions, providing a theoretical basis for the fabrication of new Pickering emulsions.
I can recommend this article, but after minor revision.
Round 2
Reviewer 3 Report
The response of the authors to my questions is not fully satisfactory.
The question is why do the authors believe that the Fuc immobilized on the Pickering emulsion droplets activate better the antioxidant enzyme system? The empirical data indicate that indeed this is so. But, in order to turn this draft into a credible scientific paper, the authors must draw a mechanistic conclusion why this is the case. For example, the authors state: "
Fuc mainly activate the antioxidant enzyme system, increase the activity of antioxidant enzymes, and reduce oxidation products. Changes in environmental con-ditions lead to differences in the molecular weight of Fuc as well as the content of sul-fates, thus resulting in differences in the antioxidant activity of the Pickering emulsions."
Please develop this thought and underline those best environmental conditions, in the context of Pickering emulsions that make Fuc perform best! If that is the case that the antioxidant activity of some compounds improve when used as emulsion stabilizers, this should be a scientific breakthrough, with impact in cosmetics industry where lotions and new products with enhanced antioxidant and radical scavanging activities should be produced.
Also, these results should be put into a larger context. Are there any other examples/reports in literature claiming that a Pickering emulsion stabilizer, increases its anti/oxidant activity?
Also for the response you added on lines 216-219, you say "polymerization" reaction. What are you polymerizing? Perhaps you meant aggregation?
The response of the authors to my 5th question is not satisfactory. The zeta potential measurement carried by DLS in such a heterogeneous mixture, emulsion oil droplets, with and without adsorbed particles, plus freely moving particles is flawed, because you simply cannot isolate the contribution of different structures present in the system to the overall observed value of the zeta potential number. In other words the conclusions are not meaningful.
lines 277-279 the authors cite the Figure 3B1-B4 as corresponding to change in NaCl concentration, yet in the figure 3 caption, the B1-B4 are described by changes in temperature and only C1-C4 due to salt concentration. Very confusing.
Figure 3C1-C4 are not cited in the text, I presume it should be cited somewhere in the lines 290-297. In fact, the entire section describing the SEM results between lines 290-297 is not credible. If the network of nanoparticles forming on the Pickering emulsion oil droplets is what should be monitoring, there is no such network forming at any of the pHs. At pH 7 we should be seeing that, because as the authors claim at lines 192-193, at pH 7 the particles are uniformly adsorbed on the emulsion droplets. Same claim in the abstract, lines 14-15, but the SEM images do not show this.
In other words the SEM images should show as clear as day and night what is claimed in the text of this paper.
I recommend the authors to really strive to make a thorough and rigurous analysis backed up by proper measurements to improve the scientific soundness of the paper.
Author Response
Manuscript ID:foods-1704708
Thank you for providing us with the reviewer’s comments on our manuscript entitled “Oppositely charged Pickering emulsion co-stabilized by chitin nanoparticles and fucoidan: Influence of environmental stresses on stability and antioxidant activity” and the opportunity to submit a revised version.
We have carefully considered your comments, as well as those of the reviewers and have amended the manuscript accordingly. Our responses to the comments provided and details of the revisions made have been provided in the following pages.
We trust that the revised manuscript meets with your approval and that it is now suitable for publication in Foods.
Thank you for your consideration. I look forward to hearing from you.
Sincerely,
Yang Li,
College of Food Science,
Northeast Agricultural University,
Harbin, Heilongjiang 150030, China.
Tel.: +86-0451-55190716
yangli@neau.edu.cn
The question is why do the authors believe that the Fuc immobilized on the Pickering emulsion droplets activate better the antioxidant enzyme system? The empirical data indicate that indeed this is so. But, in order to turn this draft into a credible scientific paper, the authors must draw a mechanistic conclusion why this is the case. For example, the authors state: "
Fuc mainly activate the antioxidant enzyme system, increase the activity of antioxidant enzymes, and reduce oxidation products. Changes in environmental conditions lead to differences in the molecular weight of Fuc as well as the content of sul-fates, thus resulting in differences in the antioxidant activity of the Pickering emulsions."
Please develop this thought and underline those best environmental conditions, in the context of Pickering emulsions that make Fuc perform best! If that is the case that the antioxidant activity of some compounds improve when used as emulsion stabilizers, this should be a scientific breakthrough, with impact in cosmetics industry where lotions and new products with enhanced antioxidant and radical scavanging activities should be produced.
Also, these results should be put into a larger context. Are there any other examples/reports in literature claiming that a Pickering emulsion stabilizer, increases its anti/oxidant activity?
Response: Thank the reviewer for the comments. the result and discussion of antioxidant activity have been modified in the revised manuscript according to the comments as follows:
The free radical scavenging rates of ABTS+, DPPH, and O2- were used to evaluate the antioxidant activity of the Pickering emulsion under different environmental stresses (Fig. 6). The free radical scavenging rates of the Pickering emulsion under the above-described environment stresses increased with increasing concentration. The free radical scavenging rate of the Pickering emulsion at pH 7 was higher than those of the other three pH values, indicating that Fuc was not destroyed at pH 7 and still exhibited antioxidant activity. The free radical scavenging rates of the Pickering emulsion under different salt concentrations decreased with increasing salt concentration, thereby destroying the interface-network structure of the oil droplets. Moreover, the complex particles aggregated and fell off the oil droplet surface, which not only destroyed the stability of the Pickering emulsion but also reduced its antioxidant activity. The free radical scavenging rate of the Pickering emulsion at −20 °C was lower than those at the other temperatures. This was because the Pickering emulsion structure was broken at −20 °C and the complex particles were mixed with oil and water, thereby reducing the contact area with ABTS+, DPPH, and O2-, which reduced the ratio of scavenging free radicals. In summary, the antioxidant activity of Pickering emulsion was the highest under pH 7 and 25°C. And the ABTS+ scavenging rate of Pickering emulsion was the highest under 0 mM salt concentration, the DPPH and O2- scavenging rate of Pickering emulsion was the highest under 100 mM salt concentration. Cao et al. also reported that Pickering emulsion with selenium nanoparticles exhibited excellent antioxidant properties [32]. Fuc mainly activate the antioxidant enzyme system, increase the activity of antioxidant enzymes, and reduce oxidation products. Changes in environmental conditions lead to differences in the molecular weight of Fuc as well as the content of sulfates, thus resulting in differences in the antioxidant activity of the Pickering emulsions. Therefore, the Fuc-ChN Pickering emulsion with excellent antioxidant activity can be used in cosmetics and food industries.
Also for the response you added on lines 216-219, you say "polymerization" reaction. What are you polymerizing? Perhaps you meant aggregation?
Response: Thank the reviewer for the question. The polymerization has been modified to aggregation in the revised manuscript.
The response of the authors to my 5th question is not satisfactory. The zeta potential measurement carried by DLS in such a heterogeneous mixture, emulsion oil droplets, with and without adsorbed particles, plus freely moving particles is flawed, because you simply cannot isolate the contribution of different structures present in the system to the overall observed value of the zeta potential number. In other words the conclusions are not meaningful.
Response: Thank to the reviewer for the comments. Emulsions are colloidal dispersions that consist of at least two immiscible fluids (normally water and oil), with one of them being dispersed in the other in the form of small droplets (Tan & McClements, 2021; Xia, Xue, & Wei, 2021). We agree with the reviewer that emulsion was a heterogeneous mixture including emulsion oil droplet, with and without adsorbed particles. But the component was separated, the emulsion was instability. the zeta potential is a physicochemical parameter that expresses the stability of delivery system. Extremely positive or negative zeta potential values cause large repulsive forces, whereas repulsion between particles with similar electric charge prevents aggregation, and accordingly ensures easy redispersion(Lunardi, Gomes, Rocha, De Tommaso, & Patience, 2021). Meanwhile, the zeta potential of Pickering emulsion was measured according to many references, for example, (Lima Cardial, Paula, Cordeiro da Silva, da Silva Barros, Richter, Sombra, et al., 2019; Ren, Chen, Zhang, Lin, Weng, Liu, et al., 2022; Song, Xiong, Shi, Yuan, & Gao, 2022; Zhu, Lu, Zhu, Zhang, & Yin, 2019).
lines 277-279 the authors cite the Figure 3B1-B4 as corresponding to change in NaCl concentration, yet in the figure 3 caption, the B1-B4 are described by changes in temperature and only C1-C4 due to salt concentration. Very confusing.
Response: Thank the reviewer for the comments. We are very apology for confusing the reviewers due to my mistakes. The figure 3 caption has been modified in the revised manuscript as follows:
Fig. 3 Cryo-scanning electron microscopy (Cryo-SEM) images of the Pickering emulsion under different environ-mental conditions. (A1–A4) temperature, (B1–B4) salt concentration, and (C1–C4) pH.
Figure 3C1-C4 are not cited in the text, I presume it should be cited somewhere in the lines 290-297. In fact, the entire section describing the SEM results between lines 290-297 is not credible. If the network of nanoparticles forming on the Pickering emulsion oil droplets is what should be monitoring, there is no such network forming at any of the pHs. At pH 7 we should be seeing that, because as the authors claim at lines 192-193, at pH 7 the particles are uniformly adsorbed on the emulsion droplets. Same claim in the abstract, lines 14-15, but the SEM images do not show this.
In other words the SEM images should show as clear as day and night what is claimed in the text of this paper.
Response: Thank to the reviewer for the comments. Figure 3C1-C4 was cited in the revised manuscript, and the SEM image of Pickering emulsion under different pH was reanalyzed in the revised manuscript as follows:
The microstructure of emulsions varies at different pH value (Fig. 3C1-C4). There were particles attach to the surface of Pickering emulsion at pH 2. And the surface of Pickering emulsion was smooth at pH 4, 7, and 9. The environment around the droplet of Pickering emulsion was different under the different pH value. There was aggregation around the Pickering emulsion at pH 2 and pH 9. But the environment around Pickering emulsion at pH 7 was network. These results were attributed to the protonation of the amino groups on the ChN and F groups, which led to greater electrostatic repulsion between the polymer chains and the thinning of the interfacial film due to particle dissolution. This led to membrane rupture, which resulted in coalescence, oiling off, and eventual demulsification [27].
References:
Lima Cardial, M. R., Paula, H. C. B., Cordeiro da Silva, R. B., da Silva Barros, J. F., Richter, A. R., Sombra, F. M., de Paula, R. C. M.. Pickering emulsions stabilized with cashew gum nanoparticles as indomethacin carrier. Int J Biol Macromol 2019, 132, 534-540.
Lunardi, C. N., Gomes, A. J., Rocha, F. S., De Tommaso, J., Patience, G. S.. Experimental methods in chemical engineering: Zeta potential. The Canadian Journal of Chemical Engineering 2021, 99(3), 627-639.
Ren, Z., Chen, Z., Zhang, Y., Lin, X., Weng, W., Liu, G., Li, B.. Characteristics of Pickering emulsions stabilized by tea water-insoluble protein nanoparticles at different pH values. Food Chemistry 2022, 375.
Song, T., Xiong, Z., Shi, T., Yuan, L., Gao, R.. Effect of glutamic acid on the preparation and characterization of Pickering emulsions stabilized by zein. Food Chemistry 2022, 366.
Tan, C., McClements, D. J.. Application of Advanced Emulsion Technology in the Food Industry: A Review and Critical Evaluation. 2021, 10(4), 812.
Xia, T., Xue, C., Wei, Z.. Physicochemical characteristics, applications and research trends of edible Pickering emulsions. Trends in Food Science & Technology 2021, 107, 1-15.
Zhu, Q., Lu, H., Zhu, J., Zhang, M., Yin, L. Development and characterization of pickering emulsion stabilized by zein/corn fiber gum (CFG) complex colloidal particles. Food Hydrocolloids 2019, 91, 204-213.
